# The critical role of optimized energy density in controlling void morphology and enhancing mechanical properties of L-PBF Ti-6Al-4V ELI alloy

Supapat Trithepchunlayakoon[1], Aung Nyein Soe[2], Atikom Sombatmai[2], Suppakrit Khrueaduangkham[2], Vorapat Trachoo[3], Dinesh Rokaya[4,5], Patcharapit Promoppatum[2]*, Viritpon Srimaneepong[1]*

1 Department of Prosthodontics, Faculty of Dentistry, Chulalongkorn University, Bangkok, Thailand, 2 Department of Mechanical Engineering, Faculty of Engineering, King Mongkut's University of Technology Thonburi, Bangkok, Thailand, 3 Department of Oral Surgery, Faculty of Dentistry, Chulalongkorn University, Bangkok, Thailand, 4 Clinical Sciences Department, College of Dentistry, Ajman University, Ajman, United Arab Emirates, 5 Center of Medical and Bio-Allied Health Sciences Research, Ajman University, Ajman, United Arab Emirates

* viritpon.s@chula.ac.th (VS); patcharapit.pro@kmutt.ac.th (PP)

## Abstract

Laser power is referred to as one of the critical process parameters governing the volumetric energy density in the Laser Powder Bed Fusion (L-PBF) process. The purpose of the study is to systematically investigate the influence of laser energy density on the void morphology, microstructure, and mechanical properties of the L-PBF printed parts which were fabricated with laser power ranging from 75 to 175 W. Comprehensive analysis of void defect was conducted by employing Archimedes' method, optical microscope (OM), and X-ray microcomputed tomography (Micro-CT). Surface quality was analyzed by surface roughness measurement. Tensile testing was performed to establish the correlation between process parameters, material microstructure, and mechanical behavior in as-built samples. Under the optimal process parameters, this work achieved a minimum void fraction of 0.3%. At various laser energy densities, three distinct morphologies, namely lack of fusion (LOF), gas pores (GP), and keyhole (KH), were generated. Notably, LOF has a more detrimental effect on tensile characteristics, in comparison to GP and KH defects if laser power was less than 100 W. Interestingly, subsurface spherical pores at the hatch border demonstrate a less substantial influence on the tensile behavior of as-built samples than LOF. The correlation analysis revealed that the presence of void defects primarily influenced strength, modulus of elasticity, and strain at break. Energy density proved to play a pivotal role in defect generation, non-equilibrium microstructure, and mechanical properties of L-PBF. Based on our findings, selecting 100 W of laser power with a speed of 1200 mm/sec could be an optimal choice for achieving a satisfactory result in as-built L-PBF part.

**Data availability statement:** All relevant data are within the manuscript.

**Funding:** The Faculty of Dentistry, Chulalongkorn University (grant no: DF65003), The National Research Council of Thailand (NRCT) (grant no: N23A650883) and National Science, Research and Innovation Fund (NSRF) Fiscal year 2025 (grant no: FRB680074/0164). The funders had no role in study design, data collection and analysis, decision to publish, or preparation of the manuscript.

**Competing interests:** The authors have declared that no competing interests exist.

## Introduction

Laser Powder Bed Fusion (L-PBF) stands out as a notable additive manufacturing method that utilizes a high-intensity laser source to selectively melt and fuse metal powder layer-by-layer until the final shape is achieved on the metal substrate. This layer-wise process has been developed to fabricate complex and customized parts with minimal material waste, overcoming some of the limitations found in conventional techniques [1]. Due to their high mechanical properties and biocompatibility, titanium alloys, specifically Ti-6Al-4V, have been used for biomedical devices, including orthopedic prostheses, maxillofacial prostheses, and dental prostheses [2,3].

However, producing high-quality Ti-6Al-4V components using L-PBF could be a challenge due to high-temperature gradients and a layer-wise approach, which are governed by various process parameters including laser power, scanning speed, layer thickness, and hatch spacing [4,5]. To understand the effect of these parameters, volumetric energy density, a measurement of the average energy input applied to metal powder by the volume of melt pool formed during scanning illustrating the relationship between the key process parameters, has been extensively studied. This parameter critically determines melt pool characteristics, defect formation, surface roughness, and overall part quality [6,7].

Among the challenges of L-PBF, the void defect is critical and sometimes inevitable, especially from an improper process parameter setting. Voids can manifest in various types including lack of fusion (LOF), gas pores (GP), and keyholes (KH). Each type has a distinct morphology related to different process parameters [8]. According to Kasperovich *et al.*, they investigated the porosity in 2D and 3D approaches and claimed that optimal process parameters can significantly reduce the void defect. They classified the void defect into two major types including defects from insufficient energy which are irregular-shape so-called lack of fusion defects and excessive energy which are near-spherical-shape called keyhole defects [9].

Although it is well known that L-PBF parts have a unique fine microstructure due to rapid melting and solidification, which makes L-PBF components possess superior mechanical strength compared to conventional techniques [10,11], void defects play a role in the physical and mechanical properties of the as-built L-PBF work [12,13]. Meng *et al.* noted that void defects significantly impact tensile strength, especially ductility. Moreover, the influence of defects can overcome the influence of microstructure when there is a high amount of void [12]. Gong *et al.* found that spherical voids might be less detrimental to mechanical properties than elongated ones; even a 1% lack of fusion can cause serious effects on tensile properties because of the sharp border and crack-like shape that become the stress concentration, which weakens the component. In contrast, less than 5% of spherical voids have only a few effects on the part [13].

Besides void defects, another notable drawback of the L-PBF process is surface roughness owing to its layer-wise nature. Parts produced using L-PBF often exhibit a coarser surface compared to conventionally produced ones. This can affect mechanical properties, fatigue strength, wear resistance, and even biocompatibility in medical applications [14,15]. Several factors contribute to this surface roughness, including

the stairstep effect, scanning track contour and overlap, balling, spattering, and adhered powder [14–18]. Furthermore, Hassanin *et al.* highlighted the interplay between porosity and surface roughness and indicated that increased porosity can exacerbate surface roughness, particularly on the top surface [15].

Many researchers have attempted to optimize its parameters and focus on achieving complete melting to reduce the defects. However, previous literature offers inconsistent views on optimal parameters. Sun *et al.* noted that increasing laser power can improve tensile strength, but further increase beyond the optimum point had minimal effect on tensile strength [19]. Buhairi *et al.* also state that the variation in optimal laser power found in their literature review ranges from 70 to 400 W [20]. Several researchers tried to create a process window to show the effect of process parameters on defect generation and use them to predict the type of void [8,21,22]. While Gong *et al.* found that porosity is not linearly related to energy density [8]. Some studies found that there is no single optimal energy density value that works for all L-PBF machines. Even the same value can result from various combinations of key parameters that cause different thermal behavior. Investigating the most suitable process parameter for each machine and application is necessary to produce a suitable workpiece and meet the standard requirements of biomedical applications.

Through a systematic experimental assessment, the current work was implemented to explore the impact of various laser energy densities on void defect generation, surface roughness, microstructure, and mechanical properties in the regime of lower laser energy inputs. Laser powers ranging from 75 to 175 W were chosen while keeping other process parameters constant. A comprehensive analysis of void defect would be carried out using Archimedes' method, optical micrographs, and micro-computed tomography. The results obtained from this study will be a fundamental understanding of correlations between defects and mechanical properties of as-built L-PBF Ti-6Al-4V alloy.

## Materials and methods

### Materials and sample preparation

Ti-6Al-4V ELI alloy powder (AP&C GE Additive, Canada), with a particle size distribution of 15–45 μm and an apparent density of 2.49 g∕cm$^3$ was used as powder feedstock. The test samples were additively fabricated following ASTM E8 with 16-mm gage length, 4-mm width, and 2-mm thickness using an L-PBF technique on a metal 3D printing machine (Truprint 1000, Trumpf, Germany). The laser for the printer is Ytterbium (Yb) fiber laser with wavelength 1,070 nm and spot size of 30 μm [23]. The details of the sample fabrication process have previously been reported in our prior research [24]. To avoid contamination, the oxygen in the build chamber was purged using a high-purity shielding argon gas with a speed of 2.5 m/s. The oxygen level was kept below 0.1% during fabrication so that it would have a lesser effect on the qualities of as-built samples [25]. To investigate the characteristics influenced by laser energy inputs, different laser powers (75, 100, 125, 150 and 175 W) were chosen to obtain different volumetric energy densities with a fixed scanning speed of 1200 mm/sec, layer thickness of 20 μm, hatch spacing of 110 μm, laser spot size of 30 μm and Chessboard scanning pattern. The volumetric energy density ($E_v$) is expressed as follows in Eq. 1 [26]:

$$E_v = \frac{P}{v \times h \times t}$$

(1)

where P is the laser power (W), v is the scanning speed (mm.s$^{-1}$), h is the hatch spacing (μm), and t is the layer thickness (μm). The volumetric energy density used was 28.41, 37.88, 47.35, 56.82, and 66.29 J/mm$^3$ according to laser power.

Following fabrication, the samples were detached from the base plate using a Wire Electrical Discharge Machine (WEDM) (Excetek NP600l, Taiwan) and subsequently cleaned with ethanol. Standard metallographic preparation was followed for sample characterization. The samples were sectioned along the build direction and hot-mounted in the epoxy resin. Then, the samples were ground with silicon carbide abrasive papers starting from No. 400–2000 on a mechanical polishing machine (Polishing Machine-Nano 2000, PACE Technologies, USA), followed by polishing with diamond suspension up to 1μm and colloidal silica suspension (Struers, OP-S). The samples were etched for 20 s at room temperature

with Kroll's reagent (2.5 mL hydrofluoric acid, 5 mL nitric acid, and 100 mL distilled water) to observe microstructures of as-built samples.

## Investigation of surface roughness

In the present study, surface roughness analysis for the top and side surfaces of each laser power group was investigated using a non-contact profilometer (Alicona, infiniteFocus SL, Austria) with 10 × magnification. Three randomly selected samples from each laser power group were measured by capturing the $2 \times 2$ mm$^2$ area of each sample. The average surface roughness was reported as mean profile roughness ($R_a$), which is the integral of the absolute value of the surface profile that deviates from the mean line for a specific length, as given by Eq. 2 [24].

$$Ra = \frac{1}{l} \int_0^l |z(x)| \, dx$$

(2)

where L is the evaluation length, $|z(x)|$ is the absolute value of the shaded area for the evaluation length, and dx is the surface profile segment.

## Oxygen, nitrogen, and hydrogen content analysis

Since chemical species can affect the part integrity for medical applications, their contents in as-built samples need to be analyzed. Therefore, the O/N/H analyzer (EMGA-830, Horiba, Japan) was used to determine the concentration of oxygen, nitrogen, and hydrogen in all sample groups with different laser powers. One sample was randomly chosen from each laser power group and cut into a rectangular specimen with dimensions of $3 \times 3 \times 2$ mm and an average weight of 0.1 g to measure the O/N/H content. The O/N/H measurement was conducted once for each sample group. Two non-dispersive infrared detectors measure oxygen as carbon monoxide and carbon dioxide during the analysis, while thermal conductivity detector measures nitrogen, and a non-dispersive infrared detector measure hydrogen as $H_2O$.

## Void defect analysis

Despite some limitations, Archimedes' method, Optical microscopy (OM), and X-ray micro-computed tomography (Micro-CT) methods are three widely used approaches for accessing void defects of L-PBF samples [27]. In the current study, a combined utilization of these methods for the quantification of the void fraction and comprehensive analysis of void morphology and void distribution were explored.

Following Archimedes' method, as-built tensile samples were weighed in air and water to calculate void fraction ($\phi_{Arch}$) (Eq. 3) using an analytical balance (METTLER TOLEDO, classic, AB104-S, Switzerland) [27,28]. The measurement was repeated five samples for each laser group. Although Archimedes' method is easy to implement, it is observed to have limitations that affect the precision of the result, especially when the defect contains unmelt powders [13].

$$\Phi_{Arch} \, (\%) = \frac{\rho_{bulk} - \rho_{sample}}{\rho_{bulk}} \times 100 = 1 - \left( \frac{m_a}{m_a - m_{water}} \right) \frac{\rho_{water}}{\rho_{bulk}} \times 100$$

(3)

Where $\rho_{sample}$ is the density of the sample, $m_a$ is the mass in air, $m_{water}$ is the mass in water, $\rho_{water}$ is the density of the water at 25°C ($\rho_{water}$ = 0.997 g/cm$^3$) [29], and $\rho_{bulk}$ is the bulk density of Ti-6Al-4V ($\rho_{bulk}$= 4.42 g/cm$^3$) [30].

For the OM method, one sample was randomly selected from each group, and sectioned into a $4 \times 4$ mm size. The sample preparation has been detailed in the previous section. Void fraction analysis was performed using five different optical micrographs taken from the polished cross-section of each group along the build orientation. Image analysis software (Image J) was used to convert a grayscale image into a binary image to separate voids from bulk metal, and a median filter with a radius of 2 pixels was applied to remove noises and contaminations originating from sample preparation. A void

fraction ($\Phi_{OM}$) was calculated by dividing the area of black pixels by the total number of pixels in binary images ([Eq. 4]) [27]. The advantage of this technique is that it reduces the amount of liquid trapped in the void defect. However, the void fraction obtained from this method is an estimated value that depends on how well the image represents the distribution of the defect [31].

$$\Phi_{OM}\,(\%\,) = \frac{A_{void}}{A_{total}} \times 100$$

(4)

Where $A_{void}$ is the area of black pixels, and $A_{total}$ is the area of the total pixels in binary images.

The In-vitro Micro-Computed Tomography (Micro-CT) (Skyscan 1173, BRUKER company, Belgium) with a voxel size of 6.05 μm was used to analyze the void defect in as-built samples. Rectangular specimens (4-mm width, 13-mm length, and 2-mm thickness) were taken from the gauge length of as-built samples. The three-dimensional model was then reconstructed by stitching together multiple two-dimensional images that were projected during analysis from multiple slices of as-built samples.The post processing of reconstructed 3D models and characterization of void defects have been detailed in [32,33]. The void fraction of Micro-CT ($\phi_{Micro-CT}$) is determined from the volume of void voxels divided by the total voxel of 3D reconstructed domains ([Eq. 5]) [27]. However, caution should be taken since the resolution of Micro-CT depends on the chosen voxel size, which limit the detection of voids smaller than three times voxel size (≈18 μm) [27,34].

$$\Phi_{Micro-CT}\,(\%\,) = \frac{V_{void}}{V_{total}} \times 100$$

(5)

where $V_{void}$ is the volume of void voxels, and $V_{total}$ is the total volume of voxels in the Micro-CT sample.

## Mechanical testing

Following the ASTM E8 standard, the tensile test was performed using a servo-hydraulic testing machine (Zwick Roell, HB 250, Germany) with a crosshead speed of 0.15 mm/min under quasi-static loading conditions, which is to keep a maximum strain rate of 0.001/S. The test was carried out in the same orientation as the building orientation and the displacement during loading was measured using a video extensometer attached to the machine. To ensure the reproducibility, the testing was repeated five times per each laser power group. The ultimate tensile strength ($\sigma_{UTS}$), 0.2% offset yield strength ($\sigma_y$), Young's modulus (E), and the strain at break ($\varepsilon$) obtained from engineering stress-strain curves were also compared against the void fraction for each laser power group to understand the influence of laser energy density.

## Microstructure and fractography

An optical microscope (OM, ZEISS Axio Imager 2, Germany) was used to investigate the void morphology and morphology of prior $\beta$ grain of as-built Ti-6Al-4V ELI printed parts while a scanning electron microscope (SEM, FEI Quanta 250, USA) was used to characterize detailed features of non-equilibrium $\alpha'$ martensite that stemmed from the rapid heating and cooling rate during the L-PBF process. The same sample configuration selected for void defect analysis was also used for the OM and SEM examinations of the microstructure. Failure of as-built samples during tensile testing can be correlated to many factors including surface roughness, void defects, and non-equilibrium microstructure [35]. To understand different modes of failure, we further investigated the fracture surfaces of broken tensile samples using SEM for each laser group.

## Results and discussion

### Investigation of surface roughness

The surface morphology of as-built samples at various laser powers is presented in [Fig 1a]. The result displays that the top surface of samples gets smoother with higher laser power due to the overlapping of scan tracks. This characteristic is supported by the result obtained from the measurement of mean profile roughness (Ra) on the top surface, as shown in

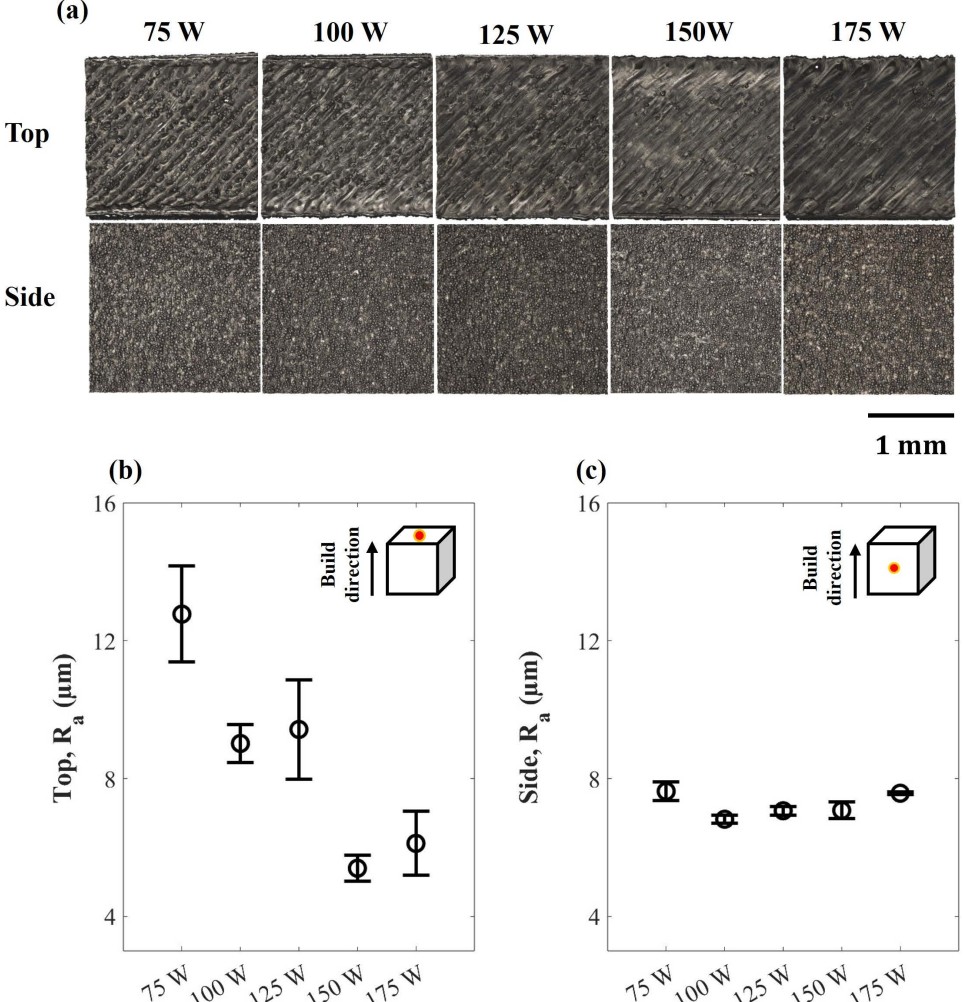

**Fig 1. (a) 3D reconstructed surface topography of all samples from the software, (b) Mean surface roughness of the top surface, and (c) Mean surface roughness of the side surface.**

Fig 1b. At 75 W, the top surface roughness is approximately 13±1 μm which is statistically significant than other groups. While 175 W reportedly achieved around 6 μm, which was 50% lower than the former. The morphology of the side surface (Fig 1c), on the other hand, differs greatly from the top surface because it dominantly consists of adhered powders. Furthermore, there is no statistically significant difference in surface roughness between all laser groups, apart from 75 W and 175 W, which have slightly higher mean roughness values of 7.6±0.3 and 7.6±0.1 μm, respectively.

The present finding demonstrates that the top surface roughness decreases with increasing laser power, which is a direct consequence of melt pool stability, greater overlapping of melting tracks, and fewer satellite particles (Fig 1a) and is consistent with Wang's study [36]. In contrast, the side surface roughness is a mix of the stair-step effect and powder-adhered roughness. This kind of roughness is often induced by powders along the lateral border of the sample that melt or partly melt because of heat radiation from the melt pool [37–39]. This study shows that increasing the laser from 75 W can reduce the surface roughness. The result corresponds with Molinari *et al.*'s work which states that increasing energy density reduces surface roughness because of the more continuous track and less adhered powder and balling

effect [17]. However, a slight increase in surface roughness was observed in 175 W, which could be attributed to several factors. First, the elevated laser power generates melt pool instabilities, leading to the formation of keyholing and spattering. These instabilities result in the formation of surface irregularities, thereby increasing roughness [40,41]. Secondly, the higher melt pool temperature can also cause more powder to adhere to the vertical side of the samples and further elevated surface roughness [37]. Moreover, Wang noted that excessive energy density can also cause over-burning and outer edge bulge phenomena, which raise the surface roughness [18].

## Oxygen, nitrogen, and hydrogen content analysis

In Table 1, the oxygen, nitrogen, and hydrogen contents in weight percentage (wt.%) of the as-built samples at different laser power groups are shown and compared with the Ti-6Al-4V ELI powder and standard ASTM data (ASTM F2924-14). The findings show that the O, N, and H contents (wt.%) of the as-built samples and Ti-6Al-4V ELI powder slightly differ, but did not exceed the ASTM standard. The increased O, N, and H contents could be due to the higher melting pool temperature by higher laser power. This could make more oxygen or nitrogen trapped during melting and solidification because titanium is known for having a strong affinity for oxygen, nitrogen, and hydrogen especially in high temperatures. Thus, this might make the melt pool susceptible to contamination with a small amount of oxygen, nitrogen, and hydrogen that existed, although the chamber environment was controlled by argon. Furthermore, surface oxidation from heat generated by high laser power causes titanium oxide on the powder surface and in the melt pool. Therefore, these oxides can be entrapped in the s olidified material, causing elevated oxygen levels [25].

## Void defect analysis

Fig 2 shows optical micrographs taken on cross-sections of as-built L-PBF samples with different laser powers. The 75 W group exhibits pores with irregular morphologies, which are commonly referred to as lack of fusion (LOF) and are likely caused by incomplete melting of powders in the melt pool during the L-PBF process. Whereas, in the samples fabricated at 100 and 125 W groups, small spherical gas pores (GP) are predominantly found. In contrast, the 150 and 175 W groups exhibit slightly different pore characteristics compared to those in the 100 and 125 W groups. The pore geometry appears to transition from round and semicircular to deep and narrow keyhole (KH) configuration, indicating deep penetration of the melt pool within alternative

The micro-CT analysis was extended to gain insight into the formation of void defects in three-dimensional space inside the as-built samples. A cuboid of $1.5 \times 1.5 \times 1.5 \, mm^3$ was extracted to display 3D void distribution and shape from all samples, including high magnification of void morphologies as shown in Fig 3. Although the presence of voids is relatively low at the laser power of 100 and 125 W, the number of voids increases when energy inputs of 75, 150, and 175 W are applied. The high magnification of pore morphology presented in Fig 3 is reasonably compatible with the observation

**Table 1. The chemical composition from O/N/H analysis (wt.%).**

| Remark | O | N | H |
|---|---|---|---|
| Ti-6Al-4V powder | 0.1012±0.0077 | 0.0234±0.0015 | 0.0033±0.0003 |
| 75 W | 0.1132±0.0010 | 0.0274±0.0007 | 0.0034±0.0005 |
| 100 W | 0.1124±0.0037 | 0.0316±0.0050 | 0.0032±0.0004 |
| 125 W | 0.1294±0.0040 | 0.0308±0.0028 | 0.0034±0.0002 |
| 150 W | 0.1189±0.0035 | 0.0298±0.0021 | 0.0031±0.0003 |
| 175 W | 0.1229±0.0116 | 0.0322±0.0086 | 0.0047±0.0003 |
| ASTM F2924-14 | 0.2 (Max) | 0.05 (Max) | 0.015(Max) |

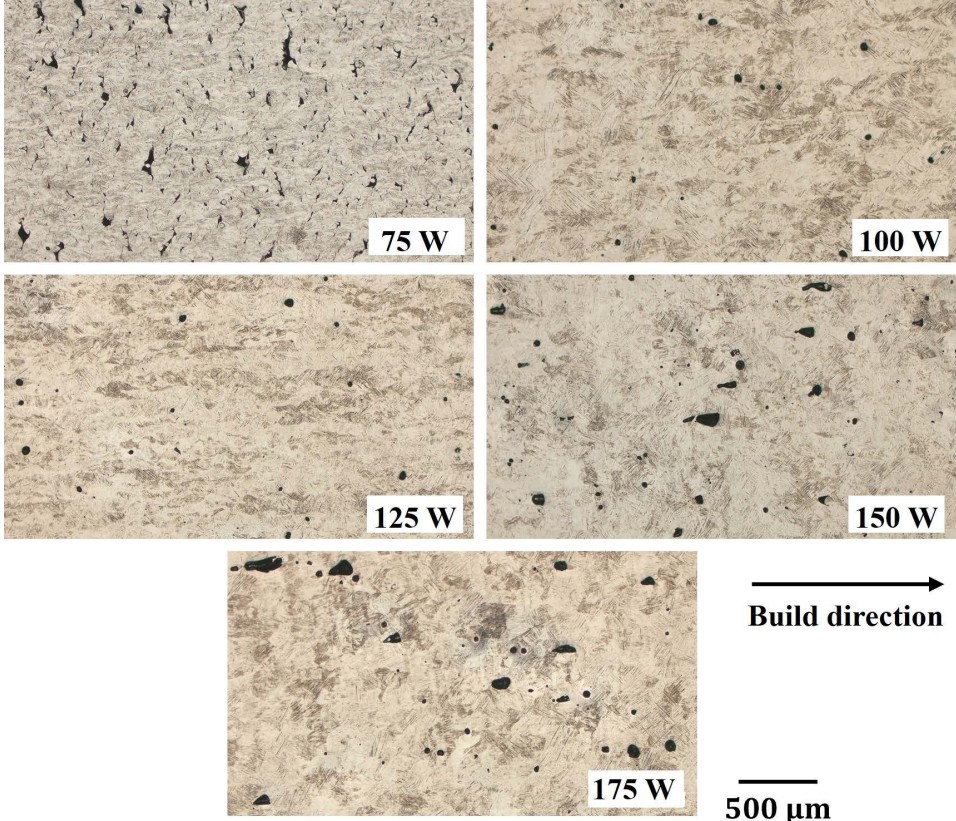

**Fig 2. Optical micrographs of as-built samples at various laser powers.** (The same images were also used to calculate void fraction for the OM method).

previously stated in the OM analysis. Unlike the OM method, the result obtained from the Micro-CT approach can depict that the mode change from GP to KH begins at 125 W and becomes more noticeable at 150 and 175 W.

In addition, as shown in both Fig 2 and Fig 3, we observed the correlation between defect characteristics and process parameters. At a low power input of 75 W, the voids appear to have irregular shapes, indicating lack of fusion defects, which result from insufficient melt pool coverage on the metal powders. At laser powers of 100 W and 125 W, the number of voids decreases, and the voids become more spherical. This observation suggests the presence of gas pores, where trapped gas is released from the powders during the melting process. Furthermore, at high laser power inputs of 150 W and 175 W, the void shapes exhibit keyholing phenomena, characterized by voids elongated along the build direction, possibly influenced by recoil pressure during molten pool vaporization [7,22].

In the current study, the comparison of void fractions using Archimedes' method, OM, and Micro-CT is shown in Fig 4a. All methods show a similar trend of the void fraction. The maximum void fraction was observed at 75 W, and as laser power increases, the void fraction decreases first before increasing after laser power exceeds 125 W. It showed that different energy inputs led to the fluctuation of void fraction in as-built L-PBF samples. This similar phenomenon has been previously described in the literature [11,19,36,42–44].

It is also observed that at 75 W, OM yields a relatively higher void fraction compared to both Micro-CT and Archimedes' methods. For energy inputs other than 75 W, the void fraction obtained through OM and Micro-CT approaches are relatively comparable. Particularly at 150 W, the result depicts a high degree of similarity across different techniques. This

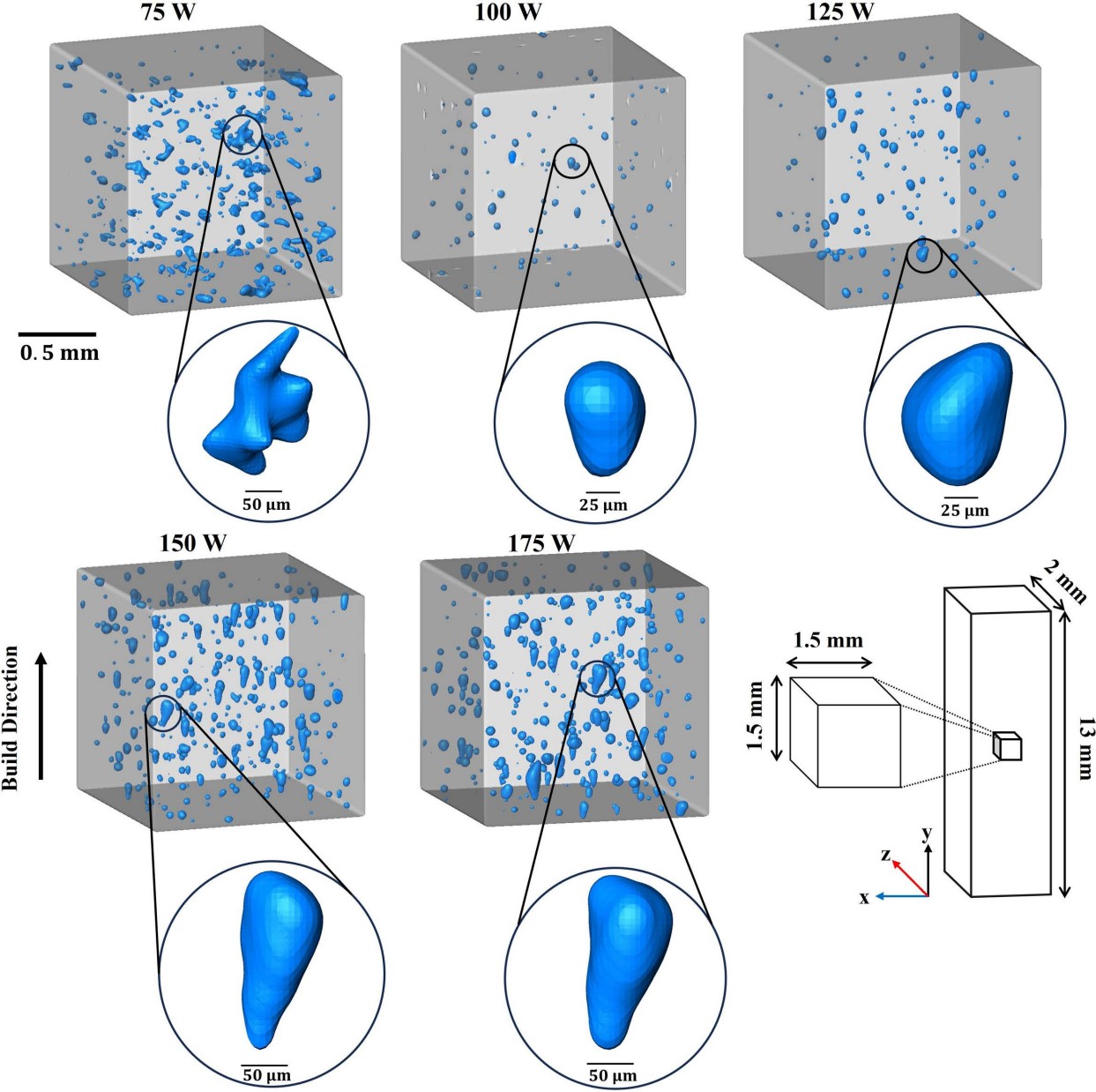

**Fig 3. Cuboid of 3D geometry reconstructed by stitching 2D images from Micro-CT analysis.** Higher magnification of void morphologies was included.

observation aligns with prior findings [33], which also highlighted the tendency of the OM method to overestimate void fractions. The Micro-CT method, on the other hand, is found to underestimate the void fraction of LOF by around 30% and 20% when compared to OM and Archimedes' methods, respectively. This could be attributed to the limitation of voxel size in this study, which fail to detect the void sizes lower than 18 μm.

Fig 4b depicts the void size distribution corresponding to different laser powers. While average 3D void sizes from Micro-CT are relatively consistent across all samples, significant deviations are observed at 75, 150, and 175

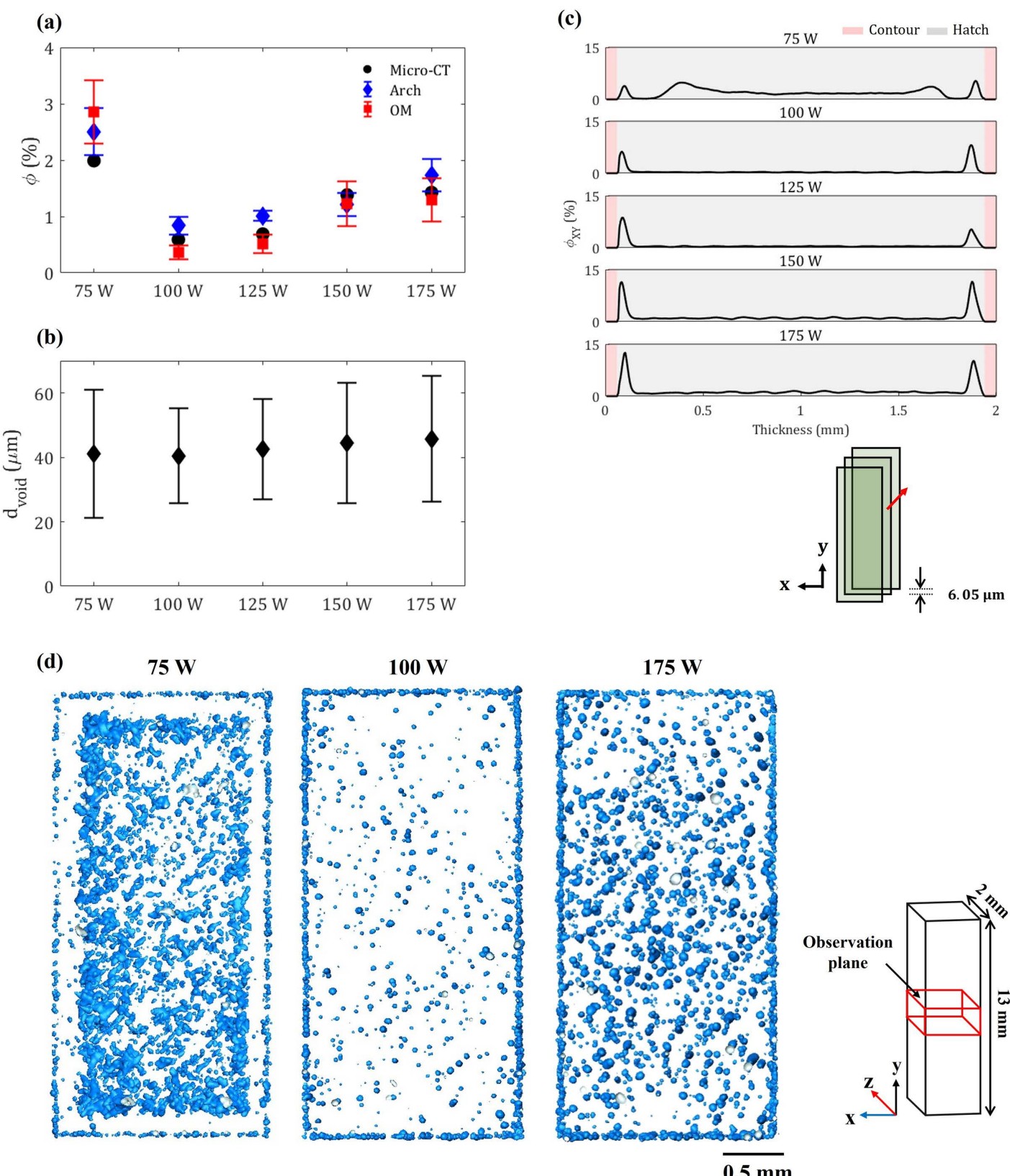

**Fig 4. (a)** Comparison of void fraction among Archimedes' method, OM, and Micro-CT approach, **(b)** Comparison of void size distribution against various laser energy inputs, **(c)** Comparison of area void fraction along the sample thickness, and **(d)** Top view of a 3D reconstructed

**domain from micro-CT at the middle region.** (Fig. 4d consists of multiple slices and were used to depict the higher pore fraction at the subsurface of as-built samples in XZ observation plane. Blue color represents void area).

**Table 2. Details on the analysis of void sizes.**

| Laser power | Numbers of voids under 1.5×1.5×1.5 mm³ sampling volume | Void size (µm) |
|---|---|---|
| 75 | 1392 | 41±20 |
| 100 | 294 | 40 ±15 |
| 125 | 322 | 42±18 |
| 150 | 636 | 42±20 |
| 175 | 524 | 46±20 |

W groups, compared to a more stable distribution at 100 and 125 W groups. At 75 W, the mean void diameter is approximately 41±20 µm, indicating considerable variability due to insufficient energy for stable melting. Similarly, at 175 W, the mean void diameter increases to 46±20 µm, suggesting that excessive energy causes melt pool instability, resulting in increasingly elongated keyholing. Similar findings on the high variability of void size distribution from Micro-CT were also observed in Promoppatum et al. [33]. Overall, the average size of void defect tends to increase slightly with higher laser power (Fig 4b), which is consistent with the study by du Plessis A [45]. In addition, the details of the analysis of void sizes and the assessment of dimensional variability of voids are shown in Table 2.

Additionally, we assessed void intensity throughout the thickness to highlight the variation in void fraction within each cross-section. Fig 4c presents the variation in area void fraction per slice. Apart from 75 W, all other groups exhibit a symmetrical void distribution pattern, characterized by higher intensity near the subsurface and lower intensity toward the center. As seen from Fig 4c, the 75 W group exhibits a unique void distribution pattern, characterized by high-intensity peaks not only at the subsurface but also at a distance of 0.2 mm from the primary peaks on both sides. The appearance of these secondary peaks of internal voids can be attributed to several possible factors. One explanation is the laser power ramping during the deposition process. During the power ramp-up phase, the laser power may fall below the threshold required for complete fusion, resulting in void formation. Alternatively, the distinct void distribution could be influenced by the thermal accumulation effect caused by the scan pattern. For instance, at the onset of laser deposition, the surrounding powder particles and the solid substrate are likely to be at a temperature close to the chamber temperature. However, as the laser continues to deposit material, it raises the temperature of the surrounding region. Consequently, the powder in subsequent laser passes may melt more effectively, leading to a non-uniform void distribution. Fig 4c also demonstrates that the peak intensity at the subsurface increases with higher laser power. Following, Fig 4d shows a top view of the 3D reconstructed void geometry in the middle region projected on the XZ plane for 75, 100, and 175 W. It is suggested that the high-intensity peak (recorded in Fig 4c) originates from a greater number of roundish voids near the end of the hatch line within the hatch region. Particularly, the morphology of voids near the surface of as-built samples appears to be consistent across all laser power groups.

Of note, although Fig 3 and Fig 4d depict void distributions from the same samples, the differences in perspective and representation may give the impression of a discrepancy. Fig 3 illustrates the void distribution within skyscan volume measuring 1.5×1.5×1.5 mm³ while Fig 4d provides a top-down view to highlight the void distribution within the XZ plane.

Martin *et al.* [46] suggested that the formation of subsurface pores in L-PBF samples is influenced by several factors such as insufficient overlapping between hatch tracks and contour tracks, laser power slowing down at the end of hatch tracks, and the laser momentarily switching off and jumping to neighboring tracks. In the current study, it could be due to

the latter that deposits small spherical pores as the laser turns around at the end of the tracks. Therefore, it is postulated that the intensity of the peak at the border of the hatch region becomes higher with increasing laser power above 75 W due to localized metal vaporization caused by higher energy density [46,47]. The presence of such subsurface pores is observed to affect the mechanical properties of L-PBF parts [48].

## Mechanical testing

The engineering stress-strain curve obtained from tensile testing is exhibited in Fig 5a. which is observed to possess good repeatability. There are statistically significant differences in yield stress, ultimate tensile stress, Young's modulus, and elongation at break among various laser powers (Fig 5b–e). According to Fig 5b and c, the lowest value of yield stress (835±25 MPa) and ultimate tensile stress (990±27 MPa) are obtained at 75 W, and their values tend to increase with increasing laser power, although above 75 W the values are statistically equivalent (Fig 5b–d). In the case of strain at break (Fig 5e), the value increases from 3.0±0.5% at 75 W to 6.6±0.5% at 100 W. After that, the value decreases with increasing laser power, 6.0±0.4% (125 W), 4.6±0.7% (150 W) and 4.0±0.5% (175 W).

Our investigation revealed that apart from the 75 W, as-built L-PBF samples consistently displayed superior yield stress and ultimate tensile stress, with average values of 1048±31 and 1187±20 MPa, respectively. These mechanical strengths are higher than ASTM standard for Additive Manufacturing Ti-6Al-4V with Powder Bed Fusion (ASTM F2924-14) which calls for minimum values of 825 and 895 MPa respectively [49]. These values are also greater compared to wrought Ti-6Al-4V ELI alloy which had yield stress and ultimate tensile stress of 945 and 979 MPa, respectively by Mower et al. [50] and 948 and 994 MPa by Shunmugavel et al. [51]. In contrast, the strain at break of the current study was lower for all energy densities (less than 7%) which is lower than wrought Ti-6Al-4V which commonly produced over 10%, 12−14% in Murr et al. [52] and 21% in Shunmugavel's work [51]. This as-built part also have less percent elongation than the minimum requirement of ASTM F2924-14 which requires at least 10% [49]. This is attributed to the characteristics of the L-PBF Ti-6Al-4V samples, which are characterized by high strength and low ductility. The high strength of L-PBF is associated with the unique fine acicular $\alpha'$ martensitic microstructure and grain refinement which mainly prevents dislocation [11,36,43,53,54]. While the formation of defects has a major role in low ductility, cause by stress concentration from the defect [19].

In addition, the tensile properties against $\Phi_{Arch}$ for each laser power is displayed in Fig 6. According to Spiering et al. [28], the Archimedes method showed higher accuracy and repeatability values than other methods, therefore, $\Phi_{Arch}$ was utilized in this part.

Fig 5 and Fig 6d show that the elongation of as-built samples is the highest at a laser power of 100 W, while the tensile strength reaches its peak at a laser power of 150 W. Interestingly, the defects in the formed parts are minimized at 100 W, yet the strength is not maximized at this laser power. This observation suggests that defects and microstructures have competing influences on mechanical properties. As discussed previously, the nature of defects transitions from lack of fusion to keyholing as laser power increases. Elongation is primarily determined by the extent of defects, as illustrated in Fig 6d. Conversely, tensile strength is influenced by additional factors, including microstructural features. Fig 7 and Fig 8 highlights the acicular features of α' martensite. According to the previous work by Promoppatum et al. [33], the size of α' martensite significantly influences the strength of printed Ti-6Al-4V. Smaller α' martensite sizes result in higher yield strength due to the effect of grain strengthening. It is anticipated that higher laser power affects the cooling rates of solidified metal, thereby inducing variations in the features of α' martensite. Supporting this, Kaschel et al. [55] reported that greater laser power produces smaller α' martensite structures. This finding aligns well with our experimental results, where the yield strength increases with higher laser power, as illustrated in Fig 5b. Overall, the mechanical properties of as-built samples are governed synergistically by defects and microstructural features, where defects primarily dictate elongation and the characteristics of microstructure, such as the size of α' martensite, play a critical role in determining tensile strength.

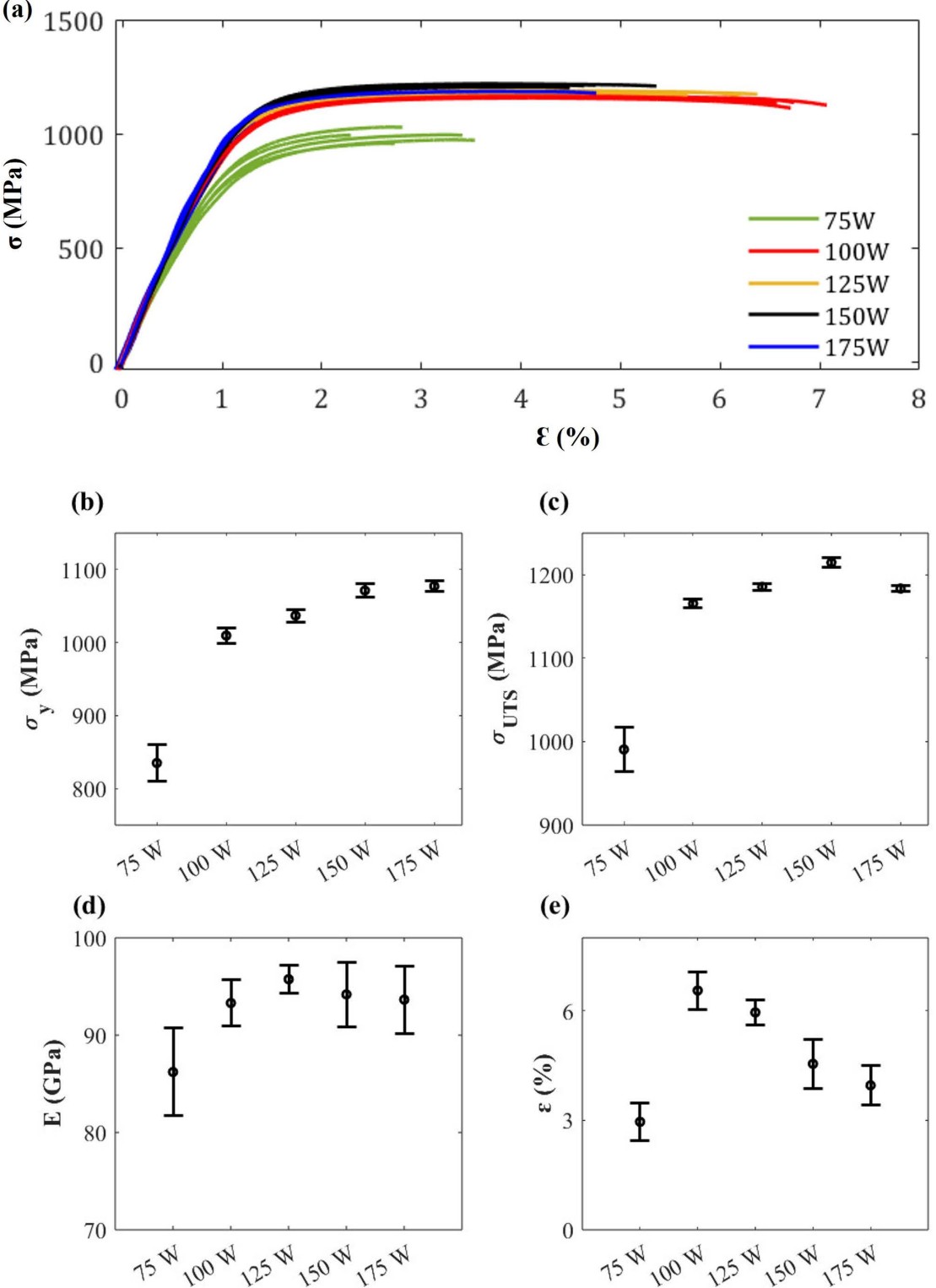

**Fig 5. (a) Engineering stress and strain curve, (b) Yield stress ($\sigma_y$),(c) Ultimate tensile stress ($\sigma_{UTS}$), (d) Young's modulus (E), and (e) Strain at break ($\varepsilon$).**

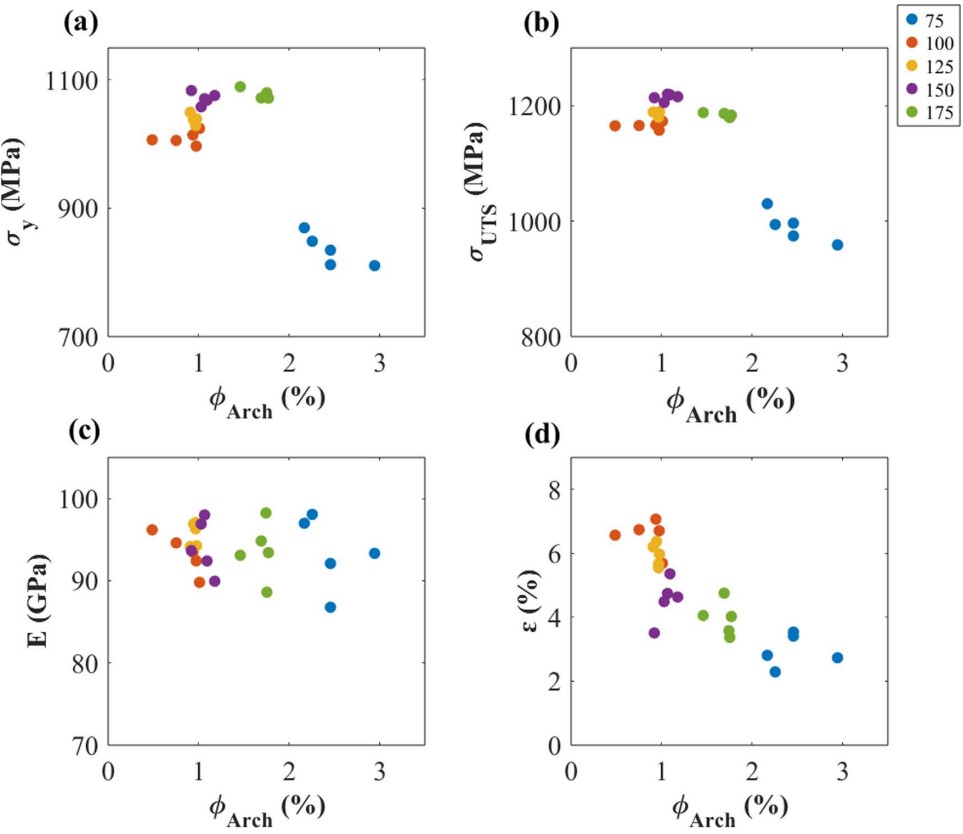

**Fig 6. Correlation between tensile properties and the void fraction from Archimedes' method.** (a) Yield stress ($\sigma_y$),(b) Ultimate tensile stress ($\sigma_{UTS}$), (c) Young's modulus (E), and (d) Strain at break ($\varepsilon$).

As previously stated, variations in laser energy inputs resulted in various modes of defects. From the tensile results, the LOF in the 75 W has a particularly detrimental effect. The irregular morphology of LOF can act as stress concentration points, initiating cracks and leading to premature fracture under tensile loading (Fig 5a). This observation is consistent with prior research highlighting the significant impact of even a small amount of LOF on the mechanical properties of L-PBF parts [9,13].

## Microstructure and fractography

The microstructure of as-built samples exhibits a fully α' martensitic microstructure for all laser power, as exhibited in Fig 7. In the current study, columnar prior $\beta$ grains found along the building direction in Fig 7 commonly occur in L-PBF due to the repeated cycle of the heating and remelting process [43]. With increasing laser power at a constant scanning speed of 1200 mm/sec, the width of columnar β grains tends to increase. For 75 W, $\beta$ grains are narrower, as shown in Fig 7a. When higher laser power was applied, the width of the prior grain size increased in Fig 7b and c. However, 150 W and 175 W caused the transition from columnar to equiaxed morphology of prior $\beta$ grains in Fig 7d and e. This characteristic is related to the excessive energy density that causes unstable melt pool and distorts the grain as reported by Han et al. [56].

Although Ti-6Al-4V is typically a dual phase $(\alpha + \beta)$ alloy, it is obvious that both OM and SEM images of as-built samples reveal fine acicular α' martensitic structure with varying sizes. Increasing laser power can produce a different

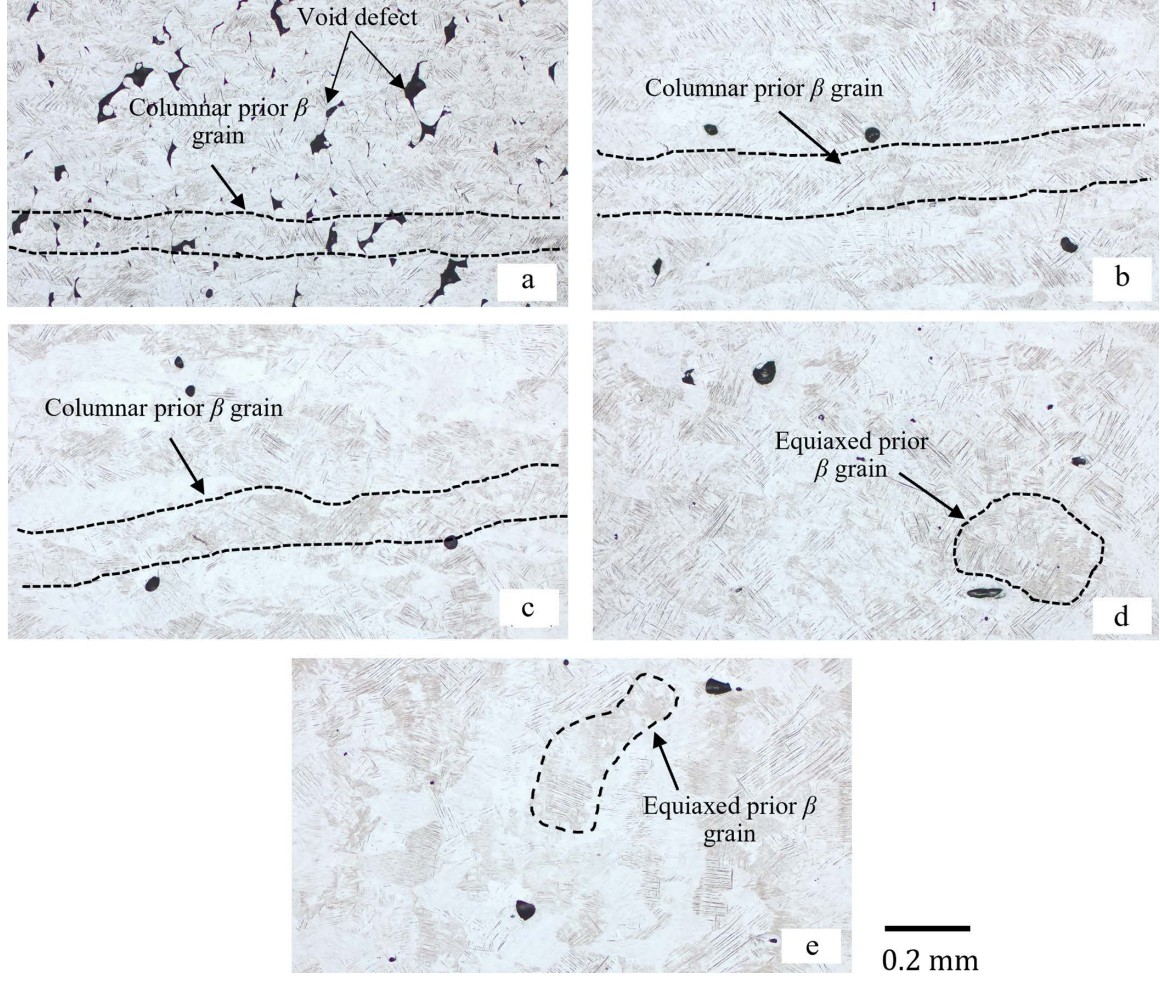

**Fig 7. Optical micrographs (OM) images of as-built samples.** (a) 75 W, (b) 100 W, (c) 125 W, (d) 150 W, and (e) 175 W.

temperature gradient in the melt pool, varying the size of the martensitic phase of each group [11], as displayed in Fig 8. Additionally, the orientations of α' grains in the microstructure presumably vary with the size of the major axis. The grains with similar sizes tend to be parallel to one another, but smaller grains indicate a tendency to orient at $45°$ to the bigger ones as shown in Fig 8. This hierarchical microstructure of acicular α' martensite is also found in Yang's work [10]. This structure can prevent the dislocation and cause poor ductility of the LPBF part [42,43].

The fracture morphology of all samples was analyzed to gain insight into the correlation between the manufacturing process and properties. Fig 9 shows the mode of failure for the respective laser groups. Following Fig 9, the middle region of the sample in each group exhibited a fibrous fracture, which is likely a ductile fracture, but the periphery of the samples displayed a thin shear lip at $45°$ along the tensile loading direction. However, each group's core area has unique features, and the width of the share lip is observed to vary in response to laser energy inputs. For 75 W, the central area has many irregular pores, uncombined regions, and unmelt powders. Additionally, the fracture surface of 75 W shows a relatively thin shear lip in comparison to other groups, indicating that the samples prematurely failed during tensile testing. Meanwhile, mixed modes of micro-dimples and cleavage fractures including a few spherical pores are typically found in the middle region of 175 W. In contrast to 75 W and 175 W, the middle area of 100 W is mainly dominated by the presence of

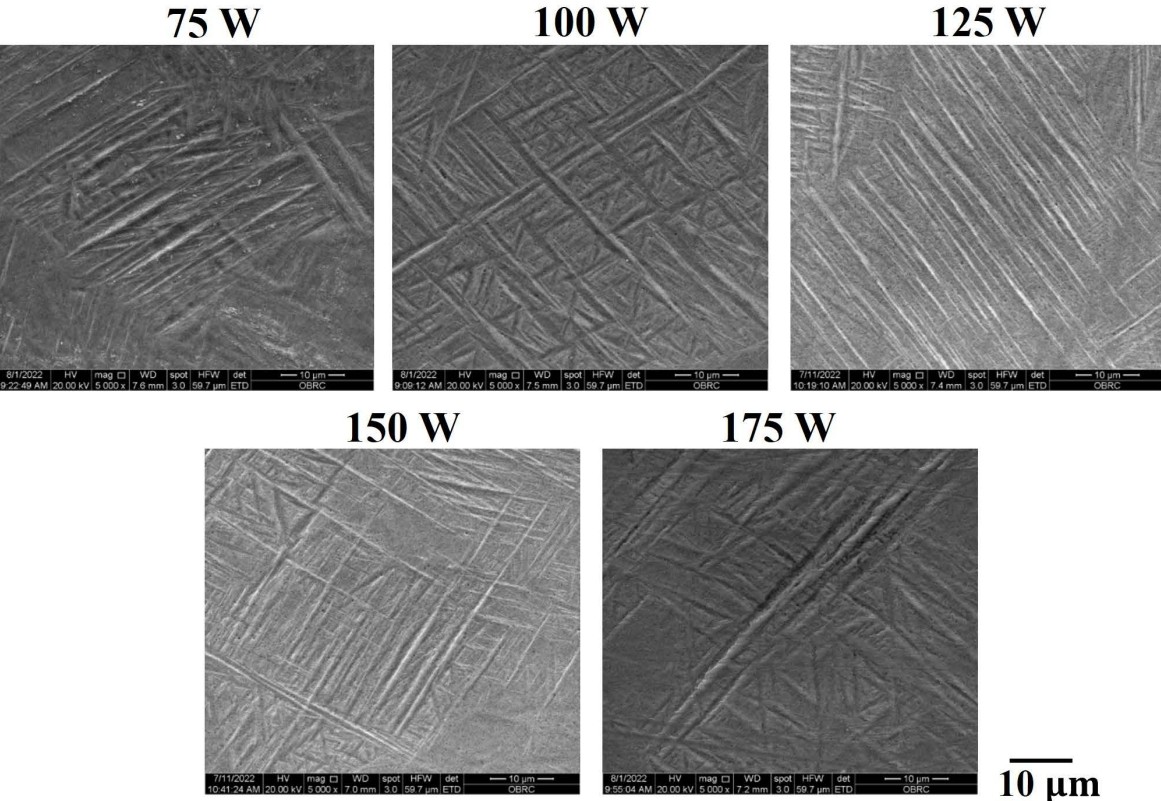

**Fig 8. Scanning electron microscopy (SEM) images of as-built samples.**

a greater number of dimple networks and a few cleavage fractures. Moreover, the shear lip is substantially broader compared to the former, reflecting the necking behavior of the samples before tensile failure.

As illustrated in Fig 9, the fracture surface of all laser powers starting from 75 to 175 W exhibits different features in the periphery and middle region. For 75 W, it is evident that the uncombined region with unmelt particles can degrade the load-bearing capacity of the cross-section during tensile loading. Additionally, the presence of small subsurface pores along the hatch region was observed, but their impact on the tensile properties of as-built L-PBF samples appeared to be less significant compared to LOF. This is typically due to their lower subsurface pore fraction at 75 W ($\sim 3.8$%) compared to 175 W ($\sim 12.5$%) (Fig 4c). Notably, in the 75 W, the width of the shear lip in the periphery region closely matched the distance between the subsurface pores region and pores intensity peak at the middle region where LOF was prevalent (Fig 4c and d), a phenomenon supported by Hu *et al.* [48]. In contrast, for other laser power groups, the ductile fracture in the central area was typically induced by the formation of the micro-voids during tensile loading. These observations underscore the critical influence of pores, particularly their void morphology, on the fracture behavior of the L-PBF samples. Furthermore, numerous micro-dimples or dimple networks were observed on the fracture surface of the same as-built samples produced at all laser power, which is an indication of plastic deformation [57,58].

## Conclusions

The present study conducted a comprehensive investigation to assess the impact of process parameters, particularly laser power, on the formation of void defects, surface quality, microstructural characteristics, and mechanical properties. The key findings from this study include:

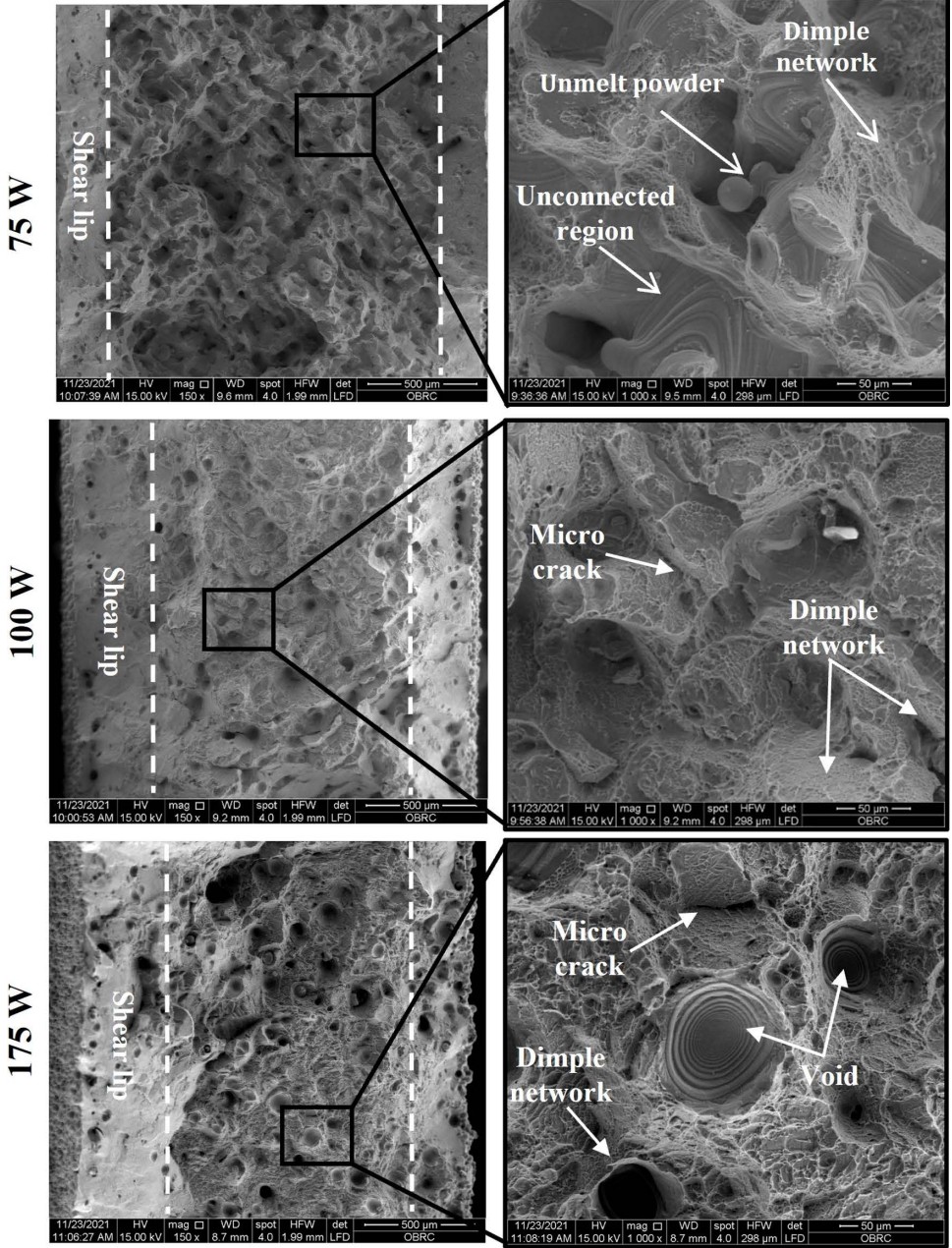

**Fig 9. Fracture surface of as-built tensile samples from 75 W, 100 W, and 175 W groups.** Higher magnification of specific areas was selected to represent the fracture behavior.

- With laser power ranging from 75 to 175 W, elongation and yield strength varied approximately from 0.05–3% and 820–1080 MPa, respectively. Selecting optimal parameters is essential to minimize defects and maximize mechanical strength.

- As energy density increased, the top surface roughness of as-built samples decreased due to improved overlap of melting tracks. Roughness decreased from approximately 12 µm to 6 µm as laser power increased from 75 to 175 W. However, side surface roughness remained relatively unchanged, mainly consisting of partially adhered powder.

- Based on void shapes, variations in energy density led to three specific types of void defects: lack of fusion (LOF), gas pores (GP), and keyhole (KH) defects. The minimum void fraction was observed at a laser power of 100 W.

- All three methods including Archimedes' method, optical microscopy (OM), and micro-computed tomography (Micro-CT) exhibited a similar trend in void fraction. However, Micro-CT specifically revealed variations in area void fraction across the sample thickness and detected sub-surface pores.

- The mechanical properties of as-built samples are influenced by both defects and microstructural features. For instance, a substantial void fraction at 75 W laser power contributed to premature fracture. At higher laser power, yield strength increased due to α' grain refinement.

## Acknowledgments

The authors appreciate the support from Dr. Ammarueda Issariyapat and Horiba (Thailand) for ONH analysis.

During the preparation of this work, the authors used ChatGPT to improve readability and language. After using this tool, the authors reviewed and edited the content as needed and took full responsibility for the content of the publication.

## Author contributions

**Conceptualization:** Patcharapit Promoppatum, Viritpon Srimaneepong.

**Data curation:** Supapat Trithepchunlayakoon, Aung Nyein Soe.

**Formal analysis:** Supapat Trithepchunlayakoon, Aung Nyein Soe.

**Funding acquisition:** Viritpon Srimaneepong.

**Investigation:** Supapat Trithepchunlayakoon.

**Methodology:** Patcharapit Promoppatum, Viritpon Srimaneepong.

**Project administration:** Patcharapit Promoppatum.

**Validation:** Atikom Sombatmai, Dinesh Rokaya.

**Visualization:** Suppakrit Khrueaduangkham, Vorapat Trachoo.

**Writing – original draft:** Supapat Trithepchunlayakoon.

**Writing – review & editing:** Aung Nyein Soe, Patcharapit Promoppatum, Viritpon Srimaneepong.

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
