## [Decision Letter · Decision Letter 0]

26 Dec 2024

PONE-D-24-50441The critical role of optimized energy density in controlling void morphology and enhancing mechanical properties of L-PBF Ti-6Al-4V ELI alloyPLOS ONE

Dear Dr. Srimaneepong,

Thank you for submitting your manuscript to PLOS ONE. After careful consideration, we feel that it has merit but does not fully meet PLOS ONE’s publication criteria as it currently stands. Therefore, we invite you to submit a revised version of the manuscript that addresses the points raised during the review process. Please, improve the description of the methods and also the discussion of the results. More detailed analysis of the results is required. More results can be extracted from microCT as pointed out by reviewers.

We look forward to receiving your revised manuscript.

Kind regards,

Antonio Riveiro Rodríguez, PhD

Academic Editor

PLOS ONE

“The Faculty of Dentistry, Chulalongkorn University (grant no: DF65003) and the National Research Council of Thailand (NRCT) (grant no: N23A650883).”

“This work was supported by the Faculty of Dentistry, Chulalongkorn University (grant no: DF65003) and the National Research Council of Thailand (NRCT) (grant no:N23A650883). The authors appreciate the support from Dr. Ammarueda Issariyapat and Horiba (Thailand) for ONH analysis. Aung Nyein Soe acknowledges the financial support from King Mongkut’s University of Technology Thonburi under the industrial Post-Doctoral and Post-Master program.”

“The Faculty of Dentistry, Chulalongkorn University (grant no: DF65003) and the National Research Council of Thailand (NRCT) (grant no: N23A650883).”

4. Please update your submission to use the PLOS LaTeX template. The template and more information on our requirements for LaTeX submissions can be found at http://journals.plos.org/plosone/s/latex.

5. In the online submission form, you indicated that [Data will be made available on request].

6. Please amend either the title on the online submission form (via Edit Submission) or the title in the manuscript so that they are identical.

Reviewers' comments:

Reviewer's Responses to Questions

**Comments to the Author**

1. Is the manuscript technically sound, and do the data support the conclusions?

Reviewer #1: Partly

Reviewer #2: No

2. Has the statistical analysis been performed appropriately and rigorously? 

Reviewer #1: Yes

Reviewer #2: No

3. Have the authors made all data underlying the findings in their manuscript fully available?

Reviewer #1: Yes

Reviewer #2: No

4. Is the manuscript presented in an intelligible fashion and written in standard English?

Reviewer #1: Yes

Reviewer #2: Yes

5. Review Comments to the Author

Reviewer #1: Paper ‘The critical role of optimized energy density in controlling void morphology and enhancing mechanical properties of L-PBF Ti-6Al-4V ELI alloy’ presents an interesting topic related to optimal processing parameters and laser additive manufacturing. The influence of laser power on defects, microstructure and mechanical behavior of L-PBF fabricated TC4 alloy was studied. There's been a lot of researches on this field, and some questions should be added to improve the quality of the manuscript.

1. The parameters that determine the density of laser energy are laser power and scanning speed, hatch spacing and layer thickness. There is no clear explanation for why only the laser power is indicated as a variable in Introduction. Also, the authors did not explain why the laser scanning speed of 1200 mm/s, layer thickness of 20 µm, hatch spacing of 110 µm, laser spot size of 30 µm a were selected in section of Materials and Methods.

2. The laser spot 30 µm was used in this paper, and it is not the same as the commonly used fiber laser with spot size of 70-100 µm. The authors should indicate the type and wavelength of the laser used in section of Materials and sample preparation.

3. The shape and size of tensile samples should give in section of Mechanical testing.

4. The oxygen, nitrogen, and hydrogen contents obviously increase with the increase of laser power. The author should explain the cause of this phenomenon.

5. In Void defect analysis, the authors only analyzed the variation trend of laser power on the morphology and size of defects. It is also important for authors to provide a specific value for the size change. Furthermore, I believe that the described defect characteristics based on CT examination are not sufficient in this paper, and that the author should provide a more detailed description of defect characteristics and the correlation between these defects and process parameters.

6. From Fig.5, the elongation of as-built samples is highest for the 100 W, and the tensile strength is highest for the 150 W. As stated in the article, the defects of the formed parts are at their lowest at 100W, however, the tensile strength is not the highest at this time. This indicates that the microstructure also has a great influence on the mechanical properties. It is necessary for the authors to explain the cause of mechanical properties change, combined with the change of microstructure as a result of laser power.

Reviewer #2: Thank you for your article. There are a number of inadequecies that will need to be addressed before this is at an acceptable quality level to be publishable.

The description of the manufacture of the samples needs significant improvement, detailing all the samples manufactured, their sizes, number off etc for each of the tests performed in their analysis.

The testing methods need to be more detailed to provide details of the sampling methods used to provide statistical robustness. Also, any stated value must have an associated error value stated.

The xCT software should be utilised in the analysis to extract useful information regarding the pores, including e.g. pore volume, pore size distributions, pore geometries (average roundness, average axis dimensions), etc. There is a lot of valuable data possible from xCT that you are not using.

There appears to be discrepancy between the results in Fig 3 and those in Fig 4. This needs to be explained.

The discussion of the results is quite limited and needs improvement, with further analysis of the data required.

Some conclusions and hypotheses are made that are unsubstantiated by the results as they stand and more detailed analysis of the results, and further sample analysis (advanced SEM techniques) will be required to be able to substantiate these.

These and more trivial errors are detailed in the attachment.

6. PLOS authors have the option to publish the peer review history of their article (what does this mean? ). If published, this will include your full peer review and any attached files.

**Do you want your identity to be public for this peer review?** For information about this choice, including consent withdrawal, please see our Privacy Policy .

Reviewer #1: **Yes: ** Pei Wei

Reviewer #2: No

---

## [Author Response · Author response to Decision Letter 0]

23 Jan 2025

Responses to the reviewers’ comments

The authors would like to thank the reviewers for their valuable comments. We have carefully considered each of the reviewers’ comments as shown in black and include our reply in blue. Please note that all the changes we have made in the revised version of the manuscript are highlighted with a yellow background and the reference numbers used in this reply refer to the bibliography in the revised manuscript.

Responses to the Reviewer # 1’s comments

Paper ‘The critical role of optimized energy density in controlling void morphology and enhancing mechanical properties of L-PBF Ti-6Al-4V ELI alloy’ presents an interesting topic related to optimal processing parameters and laser additive manufacturing. The influence of laser power on defects, microstructure and mechanical behavior of L-PBF fabricated TC4 alloy was studied. There's been a lot of researchers on this field, and some questions should be added to improve the quality of the manuscript.

1. The parameters that determine the density of laser energy are laser power and scanning speed, hatch spacing and layer thickness. There is no clear explanation for why only the laser power is indicated as a variable in Introduction. Also, the authors did not explain why the laser scanning speed of 1200 mm/s, layer thickness of 20 µm, hatch spacing of 110 µm, laser spot size of 30 µm a were selected in section of Materials and Methods.

Response: We would like to thank the reviewer for the comment. The choices of parameters used in our study are based on our pilot study and recommendation from 3D printer manufacturer. The choices of laser scanning speed, layer thickness, hatch spacing, and spot size were the default settings as recommended by machine manufacturer. To reflect the comment from the reviewer, we have added details in the revised manuscript.

Page 6 Line 141-145

2. The laser spot 30 µm was used in this paper, and it is not the same as the commonly used fiber laser with spot size of 70-100 µm. The authors should indicate the type and wavelength of the laser used in section of Materials and sample preparation.

Response: Thank you for your valuable comment. The laser fiber that this printer is this machine is Ytterbium (Yb) fiber laser with a wavelength 1,070 nm and spot size of 30 µm. Generally, the machine that utilizes the smaller spot size (30-50 µm), so-called high-resolution machines, are supposed to be used with medical and dental part which require finer details.

Reference:

Nagarajan, B.; Hu, Z.; Song, X.; Zhai, W.; Wei, J. Development of Micro Selective Laser Melting: The State of the Art and Future Perspectives. Engineering 2019, 5, 702–720.

We added more explanation about laser used in the Materials and Methods part.

The laser for the printer is Ytterbium (Yb) fibre laser with wavelength 1,070 nm and spot size of 30 µm [23].

Reference:

23. Pleass, C., & Jothi, S. (2018). Influence of powder characteristics and additive manufacturing process parameters on the microstructure and mechanical behaviour of Inconel 625 fabricated by Selective Laser Melting. Additive Manufacturing, 24, 419-431.

Page 6 Line 143-144

3. The shape and size of tensile samples should give in section of Mechanical testing.

Response: Thank you for your valuable comment. We added more details about tensile samples in materials and methods.

The sizes of tensile sample followed ASTM E8, in which the detailed dimension could be also found in our previous work of Soe et al. [24 ].

Reference:

24. Soe, A. N., Sombatmai, A., Promoppatum, P., Srimaneepong, V., Trachoo, V., & Pandee, P. (2024). Effect of post-processing treatments on surface roughness and mechanical properties of laser powder bed fusion of Ti–6Al–4V. Journal of Materials Research and Technology, 32, 3788-3803.

Page 6 Line 144-145

4. The oxygen, nitrogen, and hydrogen contents obviously increase with the increase of laser power. The author should explain the cause of this phenomenon.

Response: As recommended by the reviewer, the discussion on the increase of oxygen, nitrogen, and hydrogen contents was added in the revised manuscript.

The increased O, N, and H contents could be due to the higher melting pool temperature by higher laser power. This could make more oxygen or nitrogen trapped during melting and solidification because titanium is known for having a strong affinity for oxygen, nitrogen, and hydrogen especially in high temperatures. Thus, this might make the melt pool susceptible to contamination with a small amount of oxygen, nitrogen, and hydrogen that existed, although the chamber environment was controlled by argon. Furthermore, surface oxidation from heat generated by high laser power causes titanium oxide on the powder surface and in the melt pool. Therefore, these oxides can be entrapped in the solidified material, causing elevated oxygen levels [25].

Reference:

25. Dietrich, Kai & Diller, Johannes & Dubiez, Sophie & Bauer, Dominik & Forêt, Pierre & Witt, Gerd. (2019). The influence of oxygen on the chemical composition and mechanical properties of Ti-6Al-4V during laser powder bed fusion (L-PBF). Additive Manufacturing. 32. 100980. 10.1016/j.addma.2019.100980.

Page 13 Line 295-303

5. In Void defect analysis, the authors only analyzed the variation trend of laser power on the morphology and size of defects. It is also important for authors to provide a specific value for the size change. Furthermore, I believe that the described defect characteristics based on CT examination are not sufficient in this paper, and that the author should provide a more detailed description of defect characteristics and the correlation between these defects and process parameters.

Response: As suggested by the reviewer, we have provided a more detailed description of defect characteristics and the correlation between these defects and process parameters in the revised manuscript.

In addition, as shown in both Fig. 2 and Fig. 3, we observed the correlation between defect characteristics and process parameters. At a low power input of 75 W, the voids appear to have irregular shapes, indicating lack-of-fusion defects, which result from insufficient melt pool coverage on the metal powders. At laser powers of 100 W and 125 W, the number of voids decreases, and the voids become more spherical. This observation suggests the presence of gas pores, where trapped gas is released from the powders during the melting process. Furthermore, at high laser power inputs of 150 W and 175 W, the void shapes exhibit keyholing phenomena, characterized by voids elongated along the build direction, possibly influenced by recoil pressure during molten pool vaporization [7, 22].

References:

7. Chowdhury, S., Yadaiah, N., Prakash, C., Ramakrishna, S., Dixit, S., Gupta, L. R., & Buddhi, D. (2022). Laser powder bed fusion: a state-of-the-art review of the technology, materials, properties & defects, and numerical modelling. Journal of Materials Research and Technology, 20, 2109-2172.

22. Promoppatum, P., & Yao, S. C. (2019). Analytical evaluation of defect generation for selective laser melting of metals. The International Journal of Advanced Manufacturing Technology, 103, 1185-1198.

Page 15 Line 330-338

6. From Fig.5, the elongation of as-built samples is highest for the 100 W, and the tensile strength is highest for the 150 W. As stated in the article, the defects of the formed parts are at their lowest at 100W, however, the tensile strength is not the highest at this time. This indicates that the microstructure also has a great influence on the mechanical properties. It is necessary for the authors to explain the cause of mechanical properties change, combined with the change of microstructure as a result of laser power.

Response: As suggested by the reviewer, we have provided further discussion to highlight the competing effects between microstructures and defects on strengths and elongations.

Fig. 5 and Fig 6d shows that the elongation of as-built samples is the highest at a laser power of 100 W, while the tensile strength reaches its peak at a laser power of 150 W. Interestingly, the defects in the formed parts are minimized at 100 W, yet the strength is not maximized at this laser power. This observation suggests that defects and microstructures have competing influences on the mechanical properties. As discussed previously, the nature of defects transitions from lack-of-fusion to keyholing as laser power increases. Elongation is primarily determined by the extent of defects, as illustrated in Figure 6d. Conversely, tensile strength is influenced by additional factors, including microstructural features. Figure 7 highlights the acicular features of α' martensite. According to the previous work by Promoppatum et al. [32], the size of α' martensite significantly influences the strength of printed Ti-6Al-4V. Smaller α' martensite sizes result in higher yield strength due to the effect of grain strengthening. It is anticipated that higher laser power affects the cooling rates of solidified materials, thereby inducing variations in the features of α' martensite. Supporting this, Kaschel et al. [] reported that greater laser power produces smaller α' martensite structures. This finding aligns well with our experimental results, where the yield strength increases with higher laser power, as illustrated in Figure 5b. Overall, the mechanical properties of as-built samples are governed synergistically by defects and microstructural features, where defects primarily dictate elongation and microstructural characteristics, such as the size of α' martensite, play a critical role in determining tensile strength.

References:

32. Promoppatum, P., Chayasombat, B., Soe, A. N., Sombatmai, A., Sato, Y., Suga, T., & Tsukamoto, M. (2023). In-situ modification of thermal, microstructural, and mechanical responses by altering scan lengths in laser powder bed fusion additive manufacturing of Ti-6Al-4V. Optics & Laser Technology, 164, 109525.

Page 19-20 Line 439-457

Responses to the Reviewer # 2’s comments

Thank you for your article. There are a number of inadequacies that will need to be addressed before this is at an acceptable quality level to be publishable.

1. The description of the manufacture of the samples needs significant improvement, detailing all the samples manufactured, their sizes, number off etc for each of the tests performed in their analysis.

Response: Thank you for your valuable suggestion. We have added more details about how to prepare samples in the Materials and Methods part.

Page 6 Line 141-145

2. The testing methods need to be more detailed to provide details of the sampling methods used to provide statistical robustness. Also, any stated value must have an associated error value stated.

Response: Thank you for your valuable comment. We have included SD values in the results.

3. The xCT software should be utilised in the analysis to extract useful information regarding the pores, including e.g. pore volume, pore size distributions, pore geometries (average roundness, average axis dimensions), etc. There is a lot of valuable data possible from xCT that you are not using.

Response: As suggested by the reviewer, we included further analysis on pore volume, pore size distributions, pore geometries (average roundness, average axis dimensions) in the revised manuscript.

Page 10 Line 221-231

4. There appears to be discrepancy between the results in Fig 3 and those in Fig 4. This needs to be explained.

Response: We would like to thank the reviewer for their comments. The reviewer pointed out a discrepancy between the results presented in Figures 3 and 4. Both figures display the 3D void shapes obtained from CT analysis, and we understand that the reviewer is referring to this as the perceived discrepancy. Figure 3 illustrates the void distribution within small extracted volume measuring 1.5 x 1.5 x 1.5 mm³. Additionally, Figure 4d also presents the void distribution; however, it provides a top-down view to highlight the void distribution within the x-z plane. As a result, although Figures 3 and 4d depict void distributions from the same samples, the differences in perspective and representation may give the impression of a discrepancy. To prevent the confusion, we have added in the revised manuscript the explanation.

Of note, although Figures 3 and Figure 4d depict void distributions from the same samples, the differences in perspective and representation may give the impression of a discrepancy. Figure 3 illustrates the void distribution within small extracted volume measuring 1.5 x 1.5 x 1.5 mm³ while Figure 4d provides a top-down view to highlight the void distribution within the XZ plane.

Page 17-18 Line 393-396

5. The discussion of the results is quite limited and needs improvement, with further analysis of the data required.

Response: As suggested by the reviewer, further discussion in the results and further analysis of the data are added in the revision.

6. Some conclusions and hypotheses are made that are unsubstantiated by the results as they stand and more detailed analysis of the results, and further sample analysis (advanced SEM techniques) will be required to be able to substantiate these.

Response: As suggested by the reviewer, we revised the conclusion and ensure that they are supported by the results carried out in the present work.

The present study conducted a comprehensive investigation to assess the impact of process parameters, particularly laser power, on the formation of void defects, surface quality, microstructural characteristics, and mechanical properties. The key findings from this study include:

With laser power ranging from 75 to 175 W, elongation and yield strength varied approximately from 0.05–3% and 820–1080 MPa, respectively. Selecting optimal parameters are essential to minimize defects and maximize mechanical strength.

As energy density increased, the top surface roughness of as-built samples decreased due to improved overlap of melting tracks. Roughness decreased from approximately 12 µm to 6 µm as laser power increased from 75 to 175 W. However, side surface roughness remained relatively unchanged, mainly consisting of partially adhered powder.

Based on void shapes, variations in energy density led to three specific types of void defects: lack of fusion (LOF), gas pores (GP), and keyhole (KH) defects. The minimum void fraction was observed at a laser power of 100 W.

All three methods—Archimedes’ method, optical microscopy (OM), and micro-computed tomography (Micro-CT)—exhibited a similar trend in void fraction. However, Micro-CT specifically revealed variations in area void fraction across the sample thickness and detected sub-surface pores.

The mechanical properties of as-built samples are influenced by both defects and microstructural features. For instance, a substantial void fraction at 75 W laser power contributed to premature fracture. At higher laser power, yield strength increased due to α' grain refinement.

Page 23-24 Line 524-546

Responses to comments in the manuscript

Page 6

You need to provide a description of the manufacture of all the samples, i.e their geometries, numbers off, position on the bed relative to the recoater direction and gas flow direction (position on the bed can greatly influence quality due to weld spatter, obscuration of the laser by weld fumes, variation in bed density, variation in beam quality at extremes of bed etc).

How were the samples removed from the build plate?

Response: As suggested by the reviewer, ASTM standard used in this research was explained. The details related to the fabrication of the test coupons were already reported in our prior research. Additionally, the method used for removing the samples was also mentioned in the following paragraph.

Page 6 Line 141-145 and Page 7 Line 159-161

Page 7

How many samples were assessed? How many readings were made per sample? What sample area was used?

Response: For the surface roughness test, we utilized 3 samples from each laser power group, with measurement taken in 3 are

---

## [Decision Letter · Decision Letter 1]

12 Feb 2025

PONE-D-24-50441R1The critical role of optimized energy density in controlling void morphology and enhancing mechanical properties of L-PBF Ti-6Al-4V ELI alloyPLOS ONE

Dear Dr. Srimaneepong,

Thank you for submitting your manuscript to PLOS ONE. After careful consideration, we feel that it has merit but does not fully meet PLOS ONE’s publication criteria as it currently stands. Therefore, we invite you to submit a revised version of the manuscript that addresses the points raised during the review process. Please, address the minor comments given by reviewer 2 (please, improve the presentation of the errors) and 4.

We look forward to receiving your revised manuscript.

Kind regards,

Antonio Riveiro Rodríguez, PhD

Academic Editor

PLOS ONE

Journal Requirements:

Reviewers' comments:

Reviewer's Responses to Questions

**Comments to the Author**

1. If the authors have adequately addressed your comments raised in a previous round of review and you feel that this manuscript is now acceptable for publication, you may indicate that here to bypass the “Comments to the Author” section, enter your conflict of interest statement in the “Confidential to Editor” section, and submit your "Accept" recommendation.

Reviewer #2: All comments have been addressed

Reviewer #3: All comments have been addressed

Reviewer #4: (No Response)

2. Is the manuscript technically sound, and do the data support the conclusions?

Reviewer #2: Yes

Reviewer #3: Yes

Reviewer #4: (No Response)

3. Has the statistical analysis been performed appropriately and rigorously? 

Reviewer #2: Yes

Reviewer #3: Yes

Reviewer #4: (No Response)

4. Have the authors made all data underlying the findings in their manuscript fully available?

Reviewer #2: Yes

Reviewer #3: Yes

Reviewer #4: (No Response)

5. Is the manuscript presented in an intelligible fashion and written in standard English?

Reviewer #2: Yes

Reviewer #3: Yes

Reviewer #4: (No Response)

6. Review Comments to the Author

Reviewer #2: Errors are now provided with results, but they need to be presented to the appropriate level of precision. E.g. P28, 12.772±1.394 um, should be written as 13±1, 7.631±0.265 should be written as 7.6±0.3 etc. Please amend this throughout.

Table 1 P29 still requires an assessment of error in the wt% values.

P32 No assessment of error in the dimensions of the voids. All these dimensions require an error value. Without that you cannot assess if the changes between samples is statistically significant.

Reviewer #3: the authors have addressed all of my comments in the revised version, and I have accepted the manuscript in its current form

Reviewer #4: Manuscript Number: PONE-D-24-50441R1

Manuscript Title: The critical role of optimized energy density in controlling void morphology and enhancing mechanical properties of L-PBF Ti-6Al-4V ELI alloy

1. The abstract is perfect.

2. It is better to provide references for formulas and equations.

3. Increase the quality of Figure 1.

7. PLOS authors have the option to publish the peer review history of their article (what does this mean? ). If published, this will include your full peer review and any attached files.

**Do you want your identity to be public for this peer review?** For information about this choice, including consent withdrawal, please see our Privacy Policy .

Reviewer #2: **Yes: ** Prof Gregory J Gibbons

Reviewer #3: **Yes: ** Zameer Abbas, PhD scholar in the School of Statistics East China Normal University, Shanghai, China.

Reviewer #4: **Yes: ** Behnam Akhoundi

---

## [Author Response · Author response to Decision Letter 1]

27 Feb 2025

The authors would like to thank the reviewers’ valuable comments. We have considered each of the reviewers’ comments as shown in black and include our reply in blue. Please note that all the changes we have made in the revised version of the manuscript are highlighted with a yellow background and the reference numbers used in this reply refer to the bibliography in the revised manuscript.

Responses to the Reviewer # 2’s comments

Errors are now provided with results, but they need to be presented to the appropriate level of precision. E.g. P28, 12.772±1.394 um, should be written as 13±1, 7.631±0.265 should be written as 7.6±0.3 etc. Please amend this throughout.

We would like to thank the reviewer for the comment. We changed the decimal point as recommended by the reviewer.

Page 12 Line 262:

At 75 W, the top surface roughness is approximately 12.8 1.4 μm which is statistically significant than other groups.

Furthermore, there is no statistically significant difference in surface roughness between all laser groups, apart from 75 W and 175 W, which have slightly higher mean roughness values of 7.6 0.3 and 7.6 0.1 μm, respectively.

Table 1 P29 still requires an assessment of error in the wt% values.

For Table 1, as recommended, we analyzed and displayed the error number from three tests.

Page 13 Line 303:

Table 1. The chemical composition from O/N/H analysis (wt.%)

Remark O N H

Ti-6Al-4V powder 0.1012 ± 0.0077 0.0234 ± 0.0015 0.0033 ± 0.0003

75 W 0.1132 ± 0.0010 0.0274 ± 0.0007 0.0034 ± 0.0005

100 W 0.1124 ± 0.0037 0.0316 ± 0.0050 0.0032 ± 0.0004

125 W 0.1294 ± 0.0040 0.0308 ± 0.0028 0.0034 ± 0.0002

150 W 0.1189 ± 0.0035 0.0298 ± 0.0021 0.0031 ± 0.0003

175 W 0.1229 ± 0.0116 0.0322 ± 0.0086 0.0047 ± 0.0003

ASTM F2924-14 0.2 (Max) 0.05 (Max) 0.015(Max)

P32 No assessment of error in the dimensions of the voids. All these dimensions require an error value. Without that you cannot assess if the changes between samples is statistically significant.

We sincerely appreciate the reviewer's comments. Based on our understanding, the assessment of errors in void dimensions was already included in the manuscript. However, it may not have been stated clearly, potentially leading to ambiguity. Therefore, we have revisited and clarified the assessment of errors in void dimensions herein.

The dimensions of the voids were measured using CT data, with a sampling volume of 1.5 × 1.5 × 1.5 mm³. For samples processed at laser powers of 75, 100, 125, 150, and 175 W, the numbers of voids captured within the control volumes was 1392, 294, 322, 636 and 524, respectively. Based on these captured voids, we determined the average void size and the standard deviation of void dimensions, with the results plotted in Figure 4b.

In the revised manuscript, we have also included Table 2 providing additional details on the analysis of void sizes. The changes made in the revision are outlined below.

Page 16 Line 368:

In addition, the details of the analysis of void sizes and the assessment of dimensional variability of voids are shown in Table 2.

Table 2. Details on the analysis of void sizes

Laser power Numbers of voids under 1.5x1.5x1.5 mm³ sampling volume Void size (µm)

75 1392 41.2 ±20.0

100 294 40.2 ±15.3

125 322 41.7 ±17.7

150 636 42.1 ±20.0

175 524 45.7 ±19.5

Responses to the Reviewer # 4’s comments

4. The abstract is perfect.

We would like to thank you for comment.

5. It is better to provide references for formulas and equations.

As recommended, we provided for formulas and equations used in the manuscript. Changes in the revision could be seen below.

Page 7 Line 152:

The volumetric energy density (E_v) is expressed as follows in Eq. (1) [26]:

Page 7 Line 172:

The average surface roughness was reported as mean profile roughness (R_a), which is the integral of the absolute value of the surface profile that deviates from the mean line for a specific length, as given by Eq. (2) [24].

Page 9 Line 196:

Following Archimedes’ method, as-built tensile samples were weighed in air and water to calculate void fraction (фArch) (Eq.(3)) using an analytical balance (METTLER TOLEDO, classic, AB104-S, Switzerland) [27, 28].

Page 9 Line 212:

A void fraction (ф_OM) was calculated by dividing the area of black pixels by the total number of pixels in binary images (Eq. (4)) [27].

Page 10 Line 226:

The void fraction of Micro-CT (фMicro-CT) is determined from the volume of void voxels divided by the total voxel of 3D reconstructed domains (Eq.(5)) [27].

6. Increase the quality of Figure 1.

We improved the quality of Fig 1 and attached it in the submission of revised manuscript.

---

## [Editor Report · Decision Letter 2]

30 Apr 2025

PONE-D-24-50441R2The critical role of optimized energy density in controlling void morphology and enhancing mechanical properties of L-PBF Ti-6Al-4V ELI alloyPLOS ONE

Dear Dr. Srimaneepong,

Thank you for submitting your manuscript to PLOS ONE. After careful consideration, we feel that it has merit but does not fully meet PLOS ONE’s publication criteria as it currently stands. Therefore, we invite you to submit a revised version of the manuscript that addresses the points raised during the review process.

**Please see the appended Comments from Editorial Office.**

We look forward to receiving your revised manuscript.

Kind regards,

Antonio Riveiro Rodríguez, PhD

Academic Editor

PLOS ONE

**Journal Requirements:**

**Additional Editor Comments:**

**Comments from Editorial Office** : Please further address the most recent comments of Reviewer #2 regarding quotation of values and associated uncertainties: "Errors are now provided with results, but they need to be presented to the appropriate level of precision. E.g. P28, 12.772±1.394 um, should be written as 13±1, 7.631±0.265 should be written as 7.6±0.3 etc". Throughout the manuscript, uncertainties should be quoted to 1 significant figure and associated values quoted to this same precision.

---

## [Author Response · Author response to Decision Letter 2]

4 May 2025

We would like to thank the reviewer for the comment. We corrected the quotation of values and uncertainties as recommended by the reviewer except Table 1: Chemical composition from ONH analysis because the data has 4 decimal places which cannot be changed.

Page 12 Line 262:

At 75 W, the top surface roughness is approximately 13 ± 1 μm

Page 12 Line 267:

which have slightly higher mean roughness values of 7.6 ± 0.3 and 7.6 ± 0.1 μm, respectively.

Page 16 Line 362:

At 75 W, the mean void diameter is approximately 41 ± 20 µm, indicating considerable variability due to insufficient energy for stable melting. Similarly, at 175 W, the mean void diameter increases to 46 ± 20 µm,

Page 17 Line 372:

Laser power Numbers of voids under 1.5×1.5×1.5 mm³ sampling volume Void size (µm)

75 1392 41 ± 20

100 294 40 ± 15

125 322 42 ± 18

150 636 42 ± 20

175 524 46 ± 20

---

## [Editor Report · Decision Letter 3]

12 May 2025

The critical role of optimized energy density in controlling void morphology and enhancing mechanical properties of L-PBF Ti-6Al-4V ELI alloy

PONE-D-24-50441R3

Dear Dr. Srimaneepong,

We’re pleased to inform you that your manuscript has been judged scientifically suitable for publication and will be formally accepted for publication once it meets all outstanding technical requirements.

Kind regards,

Antonio Riveiro Rodríguez, PhD

Academic Editor

PLOS ONE
---

## [Editor Report · Acceptance letter]

PONE-D-24-50441R3

PLOS ONE

Dear Dr. Srimaneepong,

I'm pleased to inform you that your manuscript has been deemed suitable for publication in PLOS ONE. Congratulations! Your manuscript is now being handed over to our production team.

Kind regards,

on behalf of

Dr. Antonio Riveiro Rodríguez

Academic Editor

PLOS ONE